# Genetic diversity among maize (*Zea mays* L.) inbred lines adapted to Japanese climates

**Shohei Mitsuhashi**⦿*

Institute of Livestock and Grassland Science, NARO, Nasushiobara, Tochigi, Japan

* shouheim@affrc.go.jp

**Data Availability Statement:** All relevant data are within the manuscript and its Supporting Information files.

**Funding:** The funder of the study was the author's Japanese research institute and funded by a

## Abstract

Understanding the genetic diversity of inbred lines is vital for development of superior $F_1$ varieties. The present study aimed to analyze Japanese maize parental inbred lines and determine their genetic diversity for future breeding. Genetic analyses were conducted using multiple methods. Principal component analysis (PCA), phylogenetic trees, and Bayesian clustering reflected borders between heterotic groups according to the derivation of each inbred line. A self-pollinated line derived from a classic $F_1$ variety and another line from an open-pollinated population from the same derivation were classified as separate components by PCA and Bayesian clustering. The result suggests that open pollination could be essential in modern breeding. Of those classified as dent or flint based on their derivation, some had a combination of all components or clusters. Therefore, the classification of inbred lines should be based on their derivation and DNA markers. The findings will be valuable for breeding and genetic studies in Japan. Additionally, these techniques may be used to obtain a more significant number of SNPs and related phenotypic data.

## Introduction

Concerns about the long-term food supply in Japan have led the Japanese government to promote a policy regarding raising food production for self-sufficiency [1]. Japanese public sectors contribute to these efforts by breeding high-yield maize (*Zea mays* L.) varieties that are well adapted to Japanese climates.

While maize is grown as a major crop in several countries, Japan has almost entirely relied on maize imports. The current policies regarding self-sufficiency in food supply have gradually increased the area under cultivation with maize in Japan. Developing new maize $F_1$ varieties better suited to Japanese climates will increase in maize production.

Maize grown for silage and grain usage differ primarily in their ripening times. While the former is harvested at the yellow ripening stage about 40 days post-silking in the Kanto region of Japan, the latter is harvested on attaining full maturity after an additional three to four weeks. Therefore, the variety's relative maturity (RM) needs to be shorter as per the cropping system.

Understanding the genetic diversity and population structure of inbred lines is essential to development of superior $F_1$ varieties [2–4] and will enable the classification of inbred lines into heterotic groups, the selection of efficient mates or testers in $F_1$ development, and the

recurrent research budget. The funder had no role in study design, data collection and analysis, decision to publish, or preparation of the manuscript.

**Competing interests:** The author(s) have declared that no competing interests exist.

introgression of superior genes from diverse genetic resources [4–7]. Japan spans a long distance from north to south, with the northernmost region, Hokkaido, opting to grow maize varieties and inbred lines with earlier ripening adapted to cold climates. The genetic basis of these varieties has been previously reported [8] and can be potentially exploited as an essential tool for breeding early maturing varieties in warmer regions. Additionally, reports on the genetic diversity of inbred lines grown south of the Kanto region are available [9,10]. However, these varieties have not been comprehensively analyzed yet. The present study aimed to extensively study Japanese parental maize inbred lines and to analyze their genetic diversity.

## Materials and methods

### Plant materials

Table 1 lists details of the 127 inbred lines used in this study. All of these were developed in the public sector research stations, namely, Hokkaido, Miyakonojo, and Nasushiobara of NARO (The National Agriculture and Food Research Organization, Japan), or Prefectural public breeding sections in Nagano, Japan. The names of these inbred lines start with the first letter of the related locations; "Ho (Hokkaido)", "Mi (Miyakonojo)", "Na (Nasushiobara)", and "CHU (Chushin area)", respectively. "Ki" is used as the name of a breeding line in Nagano Pref. and the first letter for the Kikyogahara area in Shiojiri, Nagano Pref., but it is also the name of an inbred line at Kasetsart University in Thailand [11]. To avoid confusion, we used conservation names starting with "J" and "JC" instead of "Ki". All the lines had clear derivation and heterotic groups based on breeding history documented by the developing sectors. The inbred lines were classified into some groups based on their genetic backgrounds as described by Enoki et al. [8], and partially modified following Tamaki et al. [9]. These included flint mainly developed or derived from the European region (EF), tropical inbred lines mainly developed from hybrids for summer seeding (RD), flint mainly derived from Japanese landraces (JF), dent mainly derived from US corn-belt dent (MD), and miscellaneous origin (MIS).

**DNA preparation.** DNA was extracted as previously described by Tamaki et al. [10]. Briefly, a fresh leaf section weighing about 1g from each seedling growing in a greenhouse was cut using scissors, frozen with liquid nitrogen, and milled using 'Multi-beads shocker®' (Yasui Kikai Corporation, Osaka, Japan) thrice for 10 seconds at 1500 rpm under frozen conditions. DNA was extracted using the 'DNeasy Plant Mini Kit™' (Qiagen, Venlo, Netherlands) per the manufacturer's instructions using 100 μl of the supernatant collected post-milling. DNA concentration was measured using a 'Qubit™ 2.0 Fluorometer' and 'dsDNA HS assay kit' (Thermo Fisher Scientific, Massachusetts, USA). DNA concentrations were adjusted to 10 ng/μl for sequencing reactions.

**Genotyping.** All inbred lines were genotyped using 'Maize LD Bead chip' (Illumina Inc, San Diego, USA) containing 3,047 single-nucleotide polymorphisms (SNPs) and analyzed using the software 'GenomeStudio 2.0'. While the software allows operators to adjust settings to judge genotypes on each SNP locus manually, the authors opted to follow the automatic judgment made by the software. A custom report in Plink format using Report Wizard was generated after analysis.

Markers with more than 5% missing data and less than 3% minor allele frequency were removed by 'Plink' [12] version 1.90. Highly correlated SNPs were removed by linkage disequilibrium pruning using Variance Inflation Factor = 2, resulting in 1,007 SNPs that were analyzed further.

**Statistics and population structure analysis.** Statistical analyses were adopted to investigate the genetic distinction of the inbred lines based on the 1,007 SNP marker profiles. Population structure was estimated using principal component analysis (PCA), Bayesian clustering,

**Table 1. Parental inbred lines used in the present study.**

| Inbred line | Derivation | Group † | year | Developed in ‡ |
|---|---|---|---|---|
| Ho49 | N85×Ho4 | EF | 1995 | HARC, NARO |
| Ho87 | Astrid | EF | 2001 | HARC, NARO |
| Ho90 | Raïssa/To38 | EF | 2002 | HARC, NARO |
| Ho96 | RAA45/TH8913 | EF | 2004 | HARC, NARO |
| Ho99 | EF99-7 | EF | 2005 | HARC, NARO |
| Ho100 | EF95-8 | EF | 2006 | HARC, NARO |
| Ho120 | (302101/Ho92)/TI-024 | EF | 2009 | HARC, NARO |
| Ho119 | (599646-1/Ho87)/(Ho82/Ho87) | EF | 2009 | HARC, NARO |
| Ho121 | Blizzak/(Tiberius/Ho96)S$_1$ | EF | 2011 | HARC, NARO |
| Ho130 | (302101/Oh43Ht)/Tiberius | EF | 2011 | HARC, NARO |
| Ho128 | EF04-12 | EF | 2012 | HARC, NARO |
| Ho124 | NEF02-9 | EF | 2012 | HARC, NARO |
| Ho126 | Ho96/LG3215 | EF | 2013 | HARC, NARO |
| Ho127 | (To90/303132)/{(TI9804/Ho84)/(Ho73/Ho87)}S$_3$ | EF | 2013 | HARC, NARO |
| Ho129 | NEF07-3 | EF | 2014 | HARC, NARO |
| Ho131 | EF07-4 | EF | 2018 | HARC, NARO |
| Mi71 | RD92-9 | RD | 1998 | KARC, NARO |
| Mi62 | P3286/P3470 | RD | 1997 | KARC, NARO |
| Mi91 | RD96-12 | RD | 2002 | KARC, NARO |
| Mi93 | RD97-6 | RD | 2003 | KARC, NARO |
| Mi106 | RD98-5 | RD | 2007 | KARC, NARO |
| Na2 | Hirano | JF | 1985 | ILGS, NARO |
| Na5 | Eboshi2 | JF | 1986 | ILGS, NARO |
| Na4 | Kuma | JF | 1986 | ILGS, NARO |
| Na30 | JF2C2 | JF | 1989 | ILGS, NARO |
| Na50 | JF1C1S$_3$/Tateishi1 | JF | 1991 | ILGS, NARO |
| Na57 | {P3747/(H84/B37Ht)}/Na4 | MIS (MD*JF) | 1992 | ILGS, NARO |
| Na66 | Na1/JF2C2 | JF | 1994 | ILGS, NARO |
| Mi47 | MZ021/MZ025 | JF | 1995 | KARC, NARO |
| Na76 | JF4C2 | JF | 1997 | ILGS, NARO |
| Na79 | MZ-029/MZ-019 | JF | 1998 | ILGS, NARO |
| J1608 | MF91-11 | JF | 1999 | CAES, Nagano pref. |
| Na80 | JF5C1-46 | JF | 2000 | ILGS, NARO |
| CHU44 | MF91-8 | JF | 2002 | CAES, Nagano pref. |
| Ho95 | 94GPHA | JF | 2003 | HARC, NARO |
| Na84 | MF93-11 | JF | 2003 | ILGS, NARO |
| Na83 | MF90-12 | JF | 2003 | ILGS, NARO |
| JC-009 | SF97-2 | JF | 2004 | CAES, Nagano pref. |
| J1785 | SF95-10 | JF | 2004 | CAES, Nagano pref. |
| Mi103 | MF96-2 | JF | 2005 | KARC, NARO |
| JC-026 | NF98 | JF | 2006 | CAES, Nagano pref. |
| Na85 | JF5C2 | JF | 2006 | ILGS, NARO |
| Na89 | SF97-2 | JF | 2006 | ILGS, NARO |
| Na88 | SF96-15 | JF | 2006 | ILGS, NARO |
| Na91 | SF97- 6 | JF | 2006 | ILGS, NARO |
| CHU68 | NF98 | JF | 2007 | CAES, Nagano pref. |
| Na92 | RF99 | JF | 2007 | ILGS, NARO |

(*Continued*)

**Table 1.** (Continued)

| Inbred line | Derivation | Group † | year | Developed in ‡ |
|---|---|---|---|---|
| JC-053 | NF00-4 | JF | 2008 | CAES, Nagano pref. |
| Na93 | MC99- 6 | JF | 2008 | ILGS, NARO |
| Na94 | B73/Na28 | MIS (MD*JF) | 2008 | ILGS, NARO |
| Na95 | JF99 | JF | 2008 | ILGS, NARO |
| JC-035 | (Mi47/J1690)(J1700/J1608) | JF | 2009 | CAES, Nagano pref. |
| Na97 | JF2001dig | JF | 2009 | ILGS, NARO |
| Na101 | JF2000dig | JF | 2009 | ILGS, NARO |
| Na96 | Y02-44 | JF | 2009 | ILGS, NARO |
| Mi111 | MF02-14 | JF | 2010 | KARC, NARO |
| Na103 | SF01-3-2 | JF | 2010 | ILGS, NARO |
| Na104 | MF02-6 | JF | 2011 | ILGS, NARO |
| Ho125 | (J1785/J1711)/(Na80/Ho95) | JF | 2013 | HARC, NARO |
| Na106 | JF2004-47 | JF | 2013 | ILGS, NARO |
| Mi115 | Mi47/Na50 | JF | 2014 | KARC, NARO |
| Na111 | NFM05 | JF | 2016 | ILGS, NARO |
| Na112 | EF072-11 | MIS (EF*JF) | 2016 | ILGS, NARO |
| Na113 | Na50/Teosinte | JF | 2016 | ILGS, NARO |
| Na9 | PX77A | MD | 1986 | ILGS, NARO |
| Na7 | P3424 | MD | 1986 | ILGS, NARO |
| Na17 | G4553 | MD | 1987 | ILGS, NARO |
| Na13 | P3747 | MD | 1987 | ILGS, NARO |
| Na25 | (MS142/Mo17Ht)/P3358 | MD | 1988 | ILGS, NARO |
| Na32 | H84/H95 | MD | 1989 | ILGS, NARO |
| Na29 | H84/RB259 | MD | 1989 | ILGS, NARO |
| Na36 | (H93/Pa91)/P3358 | MD | 1989 | ILGS, NARO |
| Na38 | {(B37Ht/H84)/Mo17Ht}/H84 | MD | 1989 | ILGS, NARO |
| Na34 | P3358 | MD | 1989 | ILGS, NARO |
| Na41 | (H84/R2040)/H84 | MD | 1990 | ILGS, NARO |
| Na43 | (Pa91/H93)/P3358 | MD | 1990 | ILGS, NARO |
| Na42 | (Oh43ht/MS142)/P3358 | MD | 1990 | ILGS, NARO |
| Mi29 | P3358BC/NX220 | MD | 1991 | KARC, NARO |
| Na49 | (P3358/P3732S$_3$)/P3358 | MD | 1991 | ILGS, NARO |
| Na54 | H84/PX77A | MD | 1992 | ILGS, NARO |
| Na55 | P3747/(H84/B37Ht) | MD | 1992 | ILGS, NARO |
| Na56 | H84/P3747 | MD | 1992 | ILGS, NARO |
| Na53 | P3358/(Oh43Ht/H84) | MD | 1992 | ILGS, NARO |
| Na58 | (H84/Pa91)/R2040 | MD | 1993 | ILGS, NARO |
| Na62 | Na7/Na23 | MD | 1993 | ILGS, NARO |
| Na60 | P3352(H84/R2040) | MD | 1993 | ILGS, NARO |
| Na61 | P3358/(A509/H84) | MD | 1993 | ILGS, NARO |
| J1383 | P85264 | MD | 1994 | CAES, Nagano pref. |
| J1417 | Manitoba | MD | 1994 | CAES, Nagano pref. |
| Na65 | P3352/(H84/R2040) | MD | 1994 | ILGS, NARO |
| Ho52 | P3732 | MD | 1995 | HARC, NARO |
| Ho57 | PH4304 | MD | 1995 | HARC, NARO |
| J1539 | P85264 OPEN | MD | 1996 | CAES, Nagano pref. |
| Na74 | TX8766 | MD | 1997 | ILGS, NARO |

(*Continued*)

**Table 1.** (Continued)

| Inbred line | Derivation | Group † | year | Developed in ‡ |
|---|---|---|---|---|
| Na71 | Na7/Na23 | MD | 1997 | ILGS, NARO |
| Na69 | P3352/(H84/R2040) | MD | 1997 | ILGS, NARO |
| Na72 | P3358/(OkuduruwaseS$_2$/HiranoS$_2$) | MIS (MD*JF) | 1997 | ILGS, NARO |
| Na70 | P3358/(A509/H84) | MD | 1997 | ILGS, NARO |
| Ho68 | DK403 | MD | 1998 | HARC, NARO |
| Na78 | P3358/(P3737S$_4$/(H84/B73HtS$_4$)) | MD | 1998 | ILGS, NARO |
| Na81 | Na7/Na33 | MD | 2001 | ILGS, NARO |
| Mi83 | SD95-4 | MD | 2001 | KARC, NARO |
| Mi88 | SD95-1 | MD | 2002 | KARC, NARO |
| J1706 | (W642/Ho58)/H95rhm | MD | 2003 | CAES, Nagano pref. |
| J1707 | P3358/(OkuduruwaseS$_2$/HiranoS$_2$) | MIS (MD*JF) | 2003 | CAES, Nagano pref. |
| J1698 | MD93-6 | MD | 2004 | CAES, Nagano pref. |
| JC-002 | 96GPTI | MD | 2004 | CAES, Nagano pref. |
| Na86 | G4655 | MD | 2006 | ILGS, NARO |
| JC-014 | (J1657/Na65) (J1563/J1539) | MD | 2006 | CAES, Nagano pref. |
| Ho102 | Na7/Mi29^2 | MD | 2006 | HARC, NARO |
| Ho104 | (Ho72/Ho40)/Clarica | MD | 2007 | HARC, NARO |
| JC-036 | ED99-6 | MD | 2007 | CAES, Nagano pref. |
| JC-028 | (Mi49/J1560)(J612/J1605) | MD | 2007 | CAES, Nagano pref. |
| Ho106 | MLD99-4 | MD | 2008 | HARC, NARO |
| Ho110 | {(Ho72/Ho40)/Clarica}S$_2$/(MLD99-4)S$_2$ | MD | 2009 | HARC, NARO |
| JC-037 | ND99 | MD | 2008 | CAES, Nagano pref. |
| JC-050 | ND00 | MD | 2008 | CAES, Nagano pref. |
| JC-046 | (J1703/J1605)(Na65/J1539) | MD | 2009 | CAES, Nagano pref. |
| JC-038 | ND99-21 | MD | 2009 | CAES, Nagano pref. |
| Na98 | AD99syn1 | MD | 2009 | ILGS, NARO |
| Na99 | AD99syn1 | MD | 2009 | ILGS, NARO |
| Na100 | AD99syn1 | MD | 2009 | ILGS, NARO |
| JC-054 | (Na65/Na42)/(Ho59/Ho72) | MD | 2010 | CAES, Nagano pref. |
| JC-064 | DK567/(J1704/Mi58) | MD | 2010 | CAES, Nagano pref. |
| Na102 | AD99syn1 | MD | 2010 | ILGS, NARO |
| Na109 | AD2002 | MD | 2014 | ILGS, NARO |
| Mi29SRR | P3358BC/NX220 | MD | 2014 | KARC, NARO |

† EF, RD, JF, MD and MIS indicate flint mainly developed or derived from the European region, Japanese tropical inbred lines mainly developed from hybrids for summer seeding, Japanese flint landraces, Japanese dent mainly derived from US corn-belt dent, and miscellaneous origin, respectively.

‡ HARC, ILGS, KARC, and CAES are abbreviations for Hokkaido Agricultural Research Center, NARO, Kyushu Okinawa Agricultural Research Center, NARO, Institute of Livestock and Grassland Science, NARO, and Chushin Agricultural Experiment Station, Nagano pref., respectively.

and maximum likelihood (ML) phylogenetic analysis. PCA was performed using Plink. Bayesian clustering was conducted using ADMIXTURE [13]. We assumed K = 2–10, and the optimal K value was estimated based on cross-validation error (CVE) values calculated per the ADMIXTURE manual. An ML phylogenetic tree was constructed using MEGA X ver. 10.18 [14] according to the Tamura-Nei (1993) [15] model with 1000 bootstrap replicates. The tree was drawn using the unweighted pair group method with the arithmetic mean (UPGMA) method. The mean pairwise genetic distance of proportion (p) of nucleotide sites was calculated by MEGA X as genetic distance (GD).

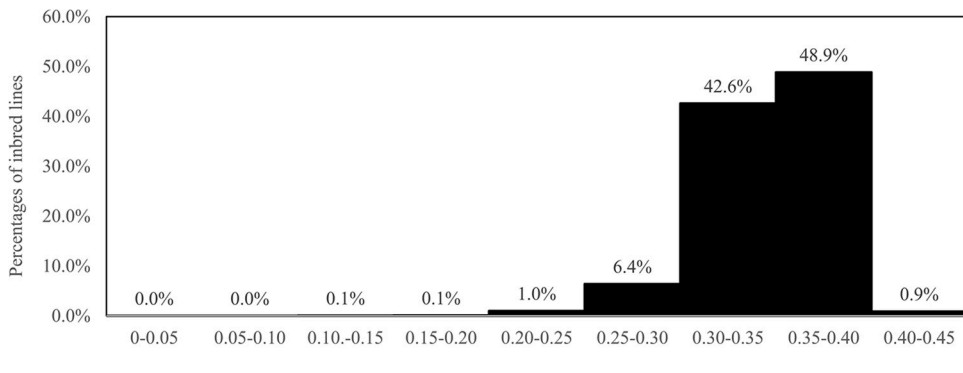

**Fig 1. Frequency distribution of pairwise genetic distances (GDs: p-distance) for 127 maize inbred lines genotyped at 1,007 SNPs.**

## Results

The frequency distribution of pairwise GDs for 127 maize inbred lines genotyped at 1,007 SNPs is shown in Fig 1. Table 2 lists the maximum, minimum, and mean GD within and among groups. S1 Table also lists all the GD matrices of 127 inbred lines. The GDs between pairwise comparisons of the inbred lines varied from 0.004 to 0.421, and the overall mean distance was 0.332. Most (48.9%) of the GDs fell between 0.350 and 0.400, with the lowest GD (0.004) being observed between 'Ho120' and 'Ho128', both of which were in the EF group but of different derivations. The highest GD of 0.421 was observed between 'Na94' and 'Ho124', which were classified with MIS (MD*JF) and EF heterotic groups and were derived from different derivations. The mean GDs between different heterotic groups tended to have low values, and minimum GDs within the same groups were relatively low, except for the RD series with lower N numbers. However, certain inbred lines had high GDs even within the same groups.

The PCA results classified EF, RD, JF and MD well (Fig 2). Inbred lines belonging to MIS, including 'Na57', 'J1707', and 'Na72' were classified near the midpoint between the JF and MD populations. Notably, 'Na94', of MIS origin of MD and JF, and 'Na112', also of EF and JF, were classified in the JF population.

The results of the population structure analysis were confirmed using a phylogenetic tree, in which the 127 genotyped inbred lines formed some groups, with each group further divided into subgroups (Fig 3). The groups agreed with the previously ascertained classification by each derivation. 'Ho95', attributed to the JF mass population, was classified to the edge of the JF group.

**Table 2. Mean genetic distances (GDs) within and between groups.**

|  | EF (N = 16) | | | RD (N = 5) | | | JF (N = 43) | | | MD (N = 63) | | |
|---|---|---|---|---|---|---|---|---|---|---|---|---|
|  | Max | Min | Mean | Max | Min | Mean | Max | Min | Mean | Max | Min | Mean |
| EF (N = 16) | 0.389 | 0.004 | 0.319 | 0.415 | 0.341 | 0.379 | 0.421 | 0.333 | 0.372 | 0.404 | 0.261 | 0.348 |
| RD (N = 5) |  |  |  | 0.320 | 0.278 | 0.301 | 0.396 | 0.313 | 0.359 | 0.398 | 0.272 | 0.352 |
| JF (N = 43) |  |  |  |  |  |  | 0.400 | 0.074 | 0.309 | 0.406 | 0.192 | 0.342 |
| MD (N = 63) |  |  |  |  |  |  |  |  |  | 0.392 | 0.012 | 0.298 |

The estimated mean GD between 127 inbred lines (GD$_M$) was 0.332.

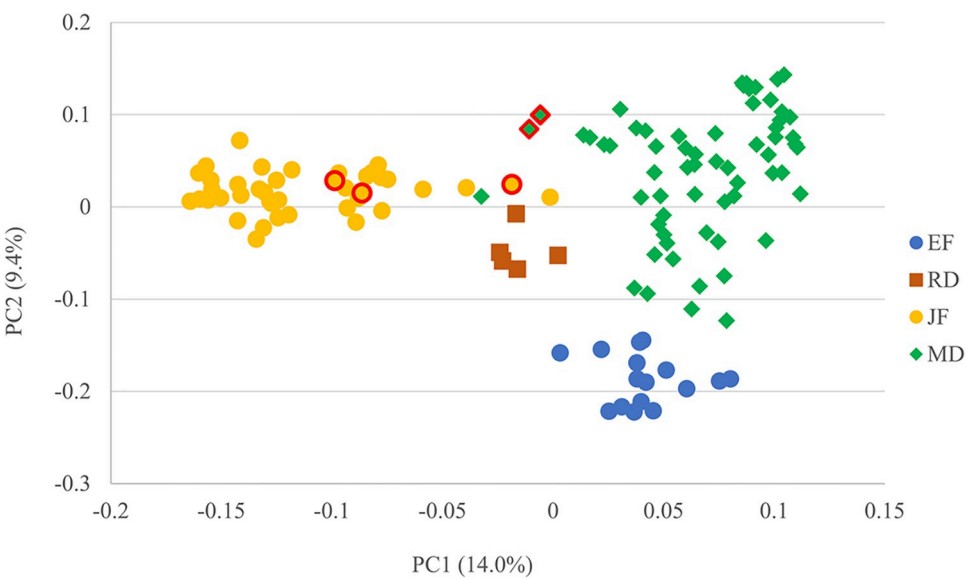

**Fig 2. A plot of principal component (PC) 1 and PC 2 scores based on 1,007 SNP markers of the 127 inbred lines.** Red line shape: Miscellaneous origin.

Fig 4 depicts Bayesian clustering by Admixture. The optimal K value was estimated to be 3 (CVE = 0.970). The second cluster was preferential in the EF group. The JF group was dominated by the first cluster and the MD group by the third, with the proportion of each cluster fluctuating as the JF and MD group approached their borders. While all clusters were approximately equal in the RD group and differed from all other heterotic groups, the proportions of the clusters were very similar to some MD inbred lines. Some of the five MIS groups and 'Ho95' contained some clusters outside the classified group. However, several inbred lines had similar clusters besides their heterotic groups.

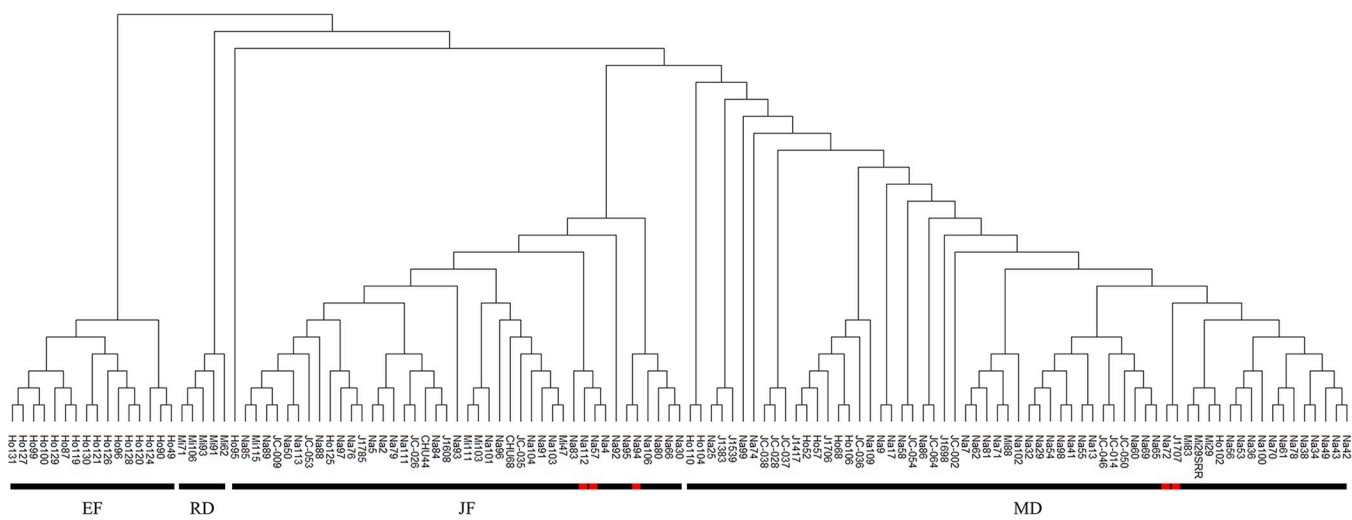

**Fig 3. The unweighted pair group method with the arithmetic mean (UPGMA) dendrogram of genetic relationships among 127 inbred lines was calculated based on genetic distances according to the Tamura-Nei model (1993).** Red shape: Miscellaneous origin.

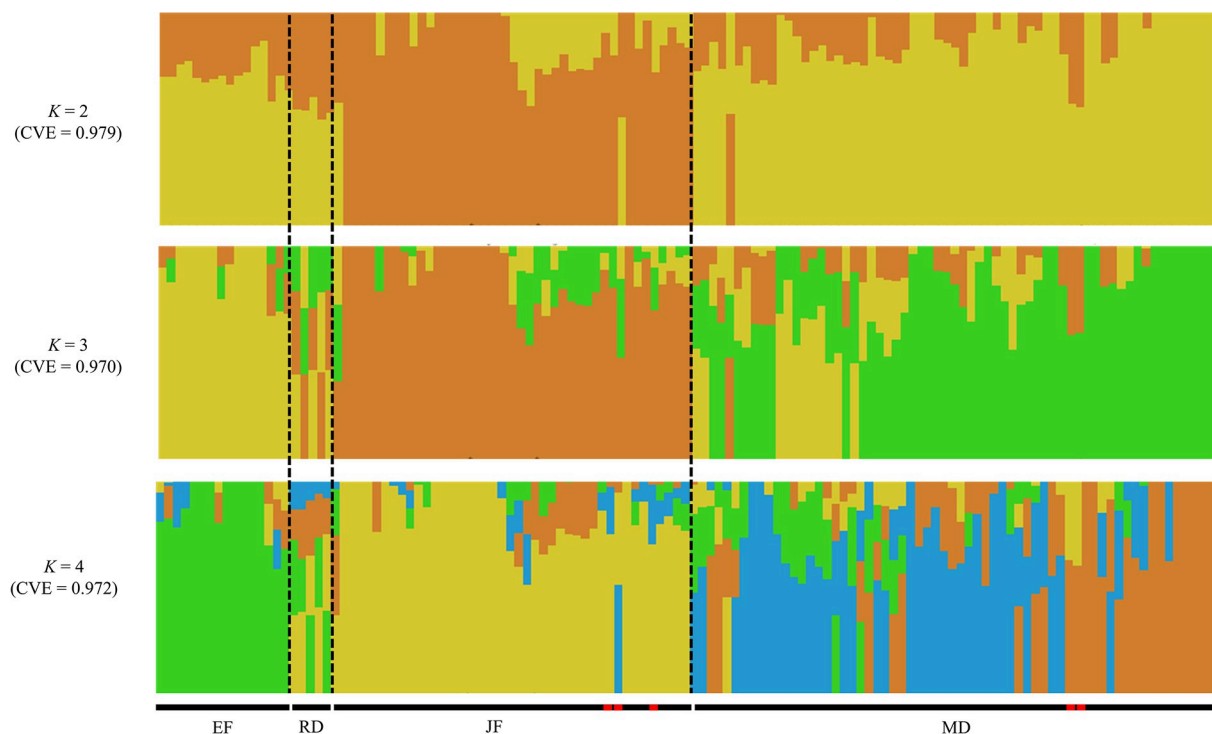

**Fig 4. ADMIXTURE model clustering from K = 2–4.** The percentage belonging to each cluster is indicated by the length of the color bar (y-axis). CVE: Cross-validation error. The sequence of each inbred line is the same as in Fig 3. Red shape: Miscellaneous origin.

## Discussion

Genetic diversity is crucial in selecting genotypes to initiate new breeding programs. New inbred lines from one group are expected to perform well in combination with inbred lines from other groups. $F_1$ hybrids result from crosses between inbred lines of different heterotic groups, the development of which is facilitated by establishing genetic similarities between inbred lines [4,5,7]. Different heterotic groups may also arise from inbred lines with a common derivation [16,17]. Genotyping is one of the most reliable approaches to documenting polymorphisms in selected inbred lines [16].

The genetic diversity of Japanese maize inbred lines was analyzed using multiple methods. PCA, phylogenetic trees, and Bayesian clustering reflected the heterotic group borders according to the derivation of each inbred line, with marginal variations due to differences in each method.

The lowest GD value was 0.004 between 'Ho120' and 'Ho128' (EF groups), derived from the triploid cross and mass selection. The GD value of 'Na89' and 'JC-009' (JF) was 0.07, derived from the same population but selected in different regions (Table 1). Although the breeding years of the two EF lines were close and the same breeders selected them, the EF lines of different origins had more significant genetic similarities than JF lines of the same origin. Warburton et al. [18] have reported that DNA markers may be better indicators of inbred lines in cases where those derived from the same population are more distinct than those derived from different populations. The findings of our study concur with this insight.

Although the average GD between different heterotic groups tended to be low, certain inbred lines had high GD even within the same group. For instance, the MD group's 'J1383' and 'J1706' had the highest GD at 0.392, with different components and clusters of structure analysis and clustering. Based on our empirical findings, $F_1$ progenies between dent lines tend

to be superior in terms of grain yield than those between dent and flint combinations, which may be an essential insight for future use of these combinations within the same group. Crosses should be made between inbred lines of different populations or the same population with high GD to ensure the development of productive $F_1$ hybrids.

PCA results indicated a clear genetic distinction between the EF, RD, JF, and MD, except in cases where certain JF and MD inbred lines were close. While 'J1539', classified with MD groups, was located close to the border with JF, 'J1383', with the same origin, was classified with MD groups. Notably, the former is derived from an open-pollinated population from a classic $F_1$ variety, and the latter, from self-pollination of itself.

Clustering analysis is the process of inferring the ancestry of inbred lines from genotype information [19]. The SNPs analyzed in this study revealed the existence of three subpopulations (K = 3) in 127 inbred lines. Inbred lines with similar derivation tended to cluster within the same group. Thus, the SNPs classified inbred lines into heterotic groups based on derivation and similar genetic backgrounds. Interestingly, 'J1383' and 'J1539', which have the same origin but a GD value of 0.303, were classified as separate components and clusters by PCA and clustering. Thus, the findings suggest that open pollination, a classic and effective tool for inducing genetic modification, could be essential in modern breeding.

'Ho95' belongs to the JF groups, which is based on the Caribbean-type flint breeding population and has a relatively later ripening period than other inbred lines bred in Hokkaido. Our previous study [10] has classified this line separate from other JF groups, which the phylogenetic tree and clustering analysis confirmed. The results of Bayesian clustering (k = 3) showed that 'Ho95' and RD groups had all clusters. Similarly, several other inbred lines had all clusters but would be classified as MD, JF, or EF group only if our recorded derivation was considered (S2 Table). 'Na112' is an MIS line derived from both EF and JF according to its derivation but had few second clusters in common with EF and a minimum GD of 0.358 from EF. Relying on these classifications will only be possible after examining them based on the origin of the underlying population and DNA markers [20]. Since the origin of these inbred lines especially before the 1990s, is based on old handwritten records, further details, including the possibility of human error, should be examined in the future.

The low cost of SNP arrays allows the analysis of numerous samples. However, given that they are developed using reference genomes, they can be confounding in diversity studies. This has been exemplified by the observation of significant confirmation bias by Ganal et al [21] using 'maize SNP50' by Illumina Inc. The present study's findings do not contradict previous study findings on genetic diversity by Rad-seq [10], and their derivations. This suggests that the SNPs array analysis used in this study helps understand genetic diversity. The development of accurate, inexpensive, and reproducible genotyping platforms has been a primary driver of genotypic studies, including those on genomic prediction [22]. The SNPs array 'Maize LD Bead chip' has already been discontinued and alternative methods will be needed in the future. Other tools for genome-wide SNP analysis should also be considered for association studies. In the future, these techniques may be used to obtain more SNPs and related phenotypic data, which could provide further insight [2,3,5,23,24].

The results of this study will serve as a valuable resource not only for maize breeding in Japan but also for genetic studies, including association mapping or genomic prediction, where genetic divergence and extended LD patterns of inbred lines are required.

## Supporting information

**S1 Table. Genetic distance matrix of 127 inbred lines in the present study.**
(XLSX)

**S2 Table. Summary of principal component and clustering using 1,007 SNP markers.**
(XLSX)

## Acknowledgments

We thank the Advanced Analysis Center of NARO, Tsukuba, Japan, for the genotyping analysis. We are also grateful to Ms. Yoko Nakada and Etsuko Imayoshi of the Institute of Livestock and Grassland Science, NARO for their assistance, and to the staff of Nasu Field Technical Team, NARO for their field management.

## Author Contributions

**Conceptualization:** Shohei Mitsuhashi.

**Data curation:** Shohei Mitsuhashi.

**Formal analysis:** Shohei Mitsuhashi.

**Funding acquisition:** Shohei Mitsuhashi.

**Investigation:** Shohei Mitsuhashi.

**Methodology:** Shohei Mitsuhashi.

**Project administration:** Shohei Mitsuhashi.

**Resources:** Shohei Mitsuhashi.

**Software:** Shohei Mitsuhashi.

**Supervision:** Shohei Mitsuhashi.

**Validation:** Shohei Mitsuhashi.

**Visualization:** Shohei Mitsuhashi.

**Writing – original draft:** Shohei Mitsuhashi.

**Writing – review & editing:** Shohei Mitsuhashi.

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
