## [Decision Letter · Decision Letter 0]

20 Oct 2023

PONE-D-23-29671Genetic diversity among maize (Zea mays L.) inbred lines adapted to Japanese climatesPLOS ONE

Dear Dr. Mitsuhashi,

Thank you for submitting your manuscript to PLOS ONE. After careful consideration, we feel that it has merit but does not fully meet PLOS ONE’s publication criteria as it currently stands. Therefore, we invite you to submit a revised version of the manuscript that addresses the points raised during the review process.

We look forward to receiving your revised manuscript.

Kind regards,

Mehdi Rahimi, Ph.D.

Academic Editor

PLOS ONE

Journal Requirements:

Reviewers' comments:

Reviewer's Responses to Questions

**Comments to the Author**

1. Is the manuscript technically sound, and do the data support the conclusions?

Reviewer #1: Yes

2. Has the statistical analysis been performed appropriately and rigorously? 

Reviewer #1: Yes

3. Have the authors made all data underlying the findings in their manuscript fully available?

Reviewer #1: Yes

4. Is the manuscript presented in an intelligible fashion and written in standard English?

Reviewer #1: Yes

5. Review Comments to the Author

Reviewer #1: This manuscript is well written.

However, I am suggesting a few corrections:

Line 21 - It is not clear what you mean by 'Self-pollinated and open-pollinated' inbred lines. Conventionally, inbred lines are developed by self pollination and maintained by self pollination. Open pollination means there is no control of the pollen parent. I have an idea what you mean by 'open pollinated inbred lines ' because I have read the whole manuscript, but in the abstract the meaning has to be very clear.

Line 74 - Change the sentence to 'DNA was extracted as previously described by Tamaki et al. [ 10 ].'

Line 199 - The phrase 'were classified as other components' may be changed to 'were classified as separate components'

6. PLOS authors have the option to publish the peer review history of their article (what does this mean?). If published, this will include your full peer review and any attached files.

Reviewer #1: No

---

## [Author Response · Author response to Decision Letter 0]

7 Nov 2023

Thank you very much for providing many useful insights. 

We have reflected these comments described below.

Reviewer #1: This manuscript is well written.

However, I am suggesting a few corrections:

Line 21 - It is not clear what you mean by 'Self-pollinated and open-pollinated' inbred lines. Conventionally, inbred lines are developed by self pollination and maintained by self pollination. Open pollination means there is no control of the pollen parent. I have an idea what you mean by 'open pollinated inbred lines ' because I have read the whole manuscript, but in the abstract the meaning has to be very clear.

　→We guess this point is very important and have corrected Abstract, according to the detailed description in the main text. 

“A self-pollinated line derived from a classic F1 variety and another line from an open-pollinated population from the same derivation were classified as separate components by PCA and Bayesian clustering.”

Line 74 - Change the sentence to 'DNA was extracted as previously described by Tamaki et al. [ 10 ].'

Line 199 - The phrase 'were classified as other components' may be changed to 'were classified as separate components'

→We modified the expression you pointed out.

We have also reflected on other points raised in another paper of ours regarding inbred names. We have also noted them in the text, 

↓(L63-)

“Ki” is used as the name of a breeding line in Nagano Pref. and the first letter for the Kikyogahara area in Shiojiri, Nagano Pref., but it is also the name of an inbred line at Kasetsart University in Thailand [11]. To avoid confusion, we have used conservation names starting with “J” and “JC” instead of “Ki”

In accordance with this description, all applicable “Ki” inbred names were corrected to the conservation number (J- or JC-) in the main text, Tables and Figures.

Table 1 etc.

We changed “KOARC, NARO” to “KARC, NARO”. It was our careless mistake.

L132-

The text was changed because the interpretation of the PCA results for 'Na94' was wrong. Also, MIS in Figure 2 is shown in red shape to make the classification visually clearer. You could have a clear picture of the three and two separated MIS.

L213-

To gain more insight into the research, we added the following description.

“Since the origin of these inbred lines especially before the 1990s, is based on old handwritten records, further details, including the possibility of human error, should be examined in the future.”

The English text has been reviewed again by a specialist and the resulting corrections have been reflected.

---

## [Decision Letter · Decision Letter 1]

23 Nov 2023

PONE-D-23-29671R1Genetic diversity among maize (Zea mays L.) inbred lines adapted to Japanese climatesPLOS ONE

Dear Dr. Mitsuhashi,

Thank you for submitting your manuscript to PLOS ONE. After careful consideration, we feel that it has merit but does not fully meet PLOS ONE’s publication criteria as it currently stands. Therefore, we invite you to submit a revised version of the manuscript that addresses the points raised during the review process.

We look forward to receiving your revised manuscript.

Kind regards,

Mehdi Rahimi, Ph.D.

Academic Editor

PLOS ONE

Journal Requirements:

Reviewers' comments:

Reviewer's Responses to Questions

**Comments to the Author**

1. If the authors have adequately addressed your comments raised in a previous round of review and you feel that this manuscript is now acceptable for publication, you may indicate that here to bypass the “Comments to the Author” section, enter your conflict of interest statement in the “Confidential to Editor” section, and submit your "Accept" recommendation.

Reviewer #1: (No Response)

2. Is the manuscript technically sound, and do the data support the conclusions?

Reviewer #1: Yes

3. Has the statistical analysis been performed appropriately and rigorously? 

Reviewer #1: Yes

4. Have the authors made all data underlying the findings in their manuscript fully available?

Reviewer #1: Yes

5. Is the manuscript presented in an intelligible fashion and written in standard English?

Reviewer #1: No

6. Review Comments to the Author

Reviewer #1: The manuscript has been improved, however, I am suggesting a few more corrections.

Line 16

The phrase ‘vital to developing’ should be changed to ‘vital for development of’

Line 32

Change ‘raising food for self-sufficiency’ to ‘raising food production for self-sufficiency’

Lines 44-46

Change sentence to ‘…. for the development of superior F1 varieties ….,. and will enable the classification of inbred lines into heterotic groups ….’

Line 63

Put a comma before the word ‘respectively’.

Line 187

Change the phrase ‘than that in’ to ‘than those between’

Line 205

Delete the words ‘that of’

7. PLOS authors have the option to publish the peer review history of their article (what does this mean?). If published, this will include your full peer review and any attached files.

Reviewer #1: No

---

## [Author Response · Author response to Decision Letter 1]

14 Dec 2023

Thank you very much again for providing many useful insights. 

We have reflected these comments described below.

Reviewer #1: The manuscript has been improved, however, I am suggesting a few more corrections.

Line 16

The phrase ‘vital to developing’ should be changed to ‘vital for development of’

Line 32

Change ‘raising food for self-sufficiency’ to ‘raising food production for self-sufficiency’

Lines 44-46

Change sentence to ‘…. for the development of superior F1 varieties ….,. and will enable the classification of inbred lines into heterotic groups ….’

Line 63

Put a comma before the word ‘respectively’.

Line 187

Change the phrase ‘than that in’ to ‘than those between’

Line 205

Delete the words ‘that of’

→We modified all the expressions you pointed out.

Also, in the tables, all relevant 'Ki' inbred names that were lost to change were corrected to their conservation numbers (J- or JC-).

And other minor corrections were made.

Journal Requirements:

→We have confirmed the reference list carefully.

---

## [Decision Letter · Decision Letter 2]

9 Jan 2024

Genetic diversity among maize (Zea mays L.) inbred lines adapted to Japanese climates

PONE-D-23-29671R2

Dear Dr. Mitsuhashi,

We’re pleased to inform you that your manuscript has been judged scientifically suitable for publication and will be formally accepted for publication once it meets all outstanding technical requirements.

Kind regards,

Mehdi Rahimi, Ph.D.

Academic Editor

PLOS ONE

Additional Editor Comments (optional):

Reviewers' comments:

Reviewer's Responses to Questions

**Comments to the Author**

1. If the authors have adequately addressed your comments raised in a previous round of review and you feel that this manuscript is now acceptable for publication, you may indicate that here to bypass the “Comments to the Author” section, enter your conflict of interest statement in the “Confidential to Editor” section, and submit your "Accept" recommendation.

Reviewer #1: All comments have been addressed

2. Is the manuscript technically sound, and do the data support the conclusions?

Reviewer #1: Yes

3. Has the statistical analysis been performed appropriately and rigorously? 

Reviewer #1: Yes

4. Have the authors made all data underlying the findings in their manuscript fully available?

Reviewer #1: Yes

5. Is the manuscript presented in an intelligible fashion and written in standard English?

Reviewer #1: Yes

6. Review Comments to the Author

Reviewer #1: Line 95

The word 'further' at the beginning of the sentence may be deleted. The sentence may start as follows: " Highly correlated....'. This is to avoid the repetition of the word 'further' in one sentence.

Try also to avoid repetition of the word 'using' in line 106.

7. PLOS authors have the option to publish the peer review history of their article (what does this mean?). If published, this will include your full peer review and any attached files.

Reviewer #1: No

---

## [Editor Report · Acceptance letter]

17 Jan 2024

PONE-D-23-29671R2 

PLOS ONE

Dear Dr. Mitsuhashi, 

I'm pleased to inform you that your manuscript has been deemed suitable for publication in PLOS ONE. Congratulations! Your manuscript is now being handed over to our production team.

Kind regards, 

on behalf of

Associate Prof. Mehdi Rahimi 

Academic Editor

PLOS ONE